# Coverage, delivery models, and implementation challenges of the community driven nutritional supplementation initiative for people with TB: A mixed methods study from Puducherry, India

Revadi Gouroumourty[1]*, Kibballi Madhukeshwar Akshaya[2], Madhur Verma[3], Premanandh Kandasamy[1], Amol Dongre[1], Murugan Natarajan[4], Swetha Mathivanan[1], Shephrine Wrobel R. Prasad[1], Hemant Deepak Shewade[5], Venkatesh Couppoussamy[6]

**1** Department of Community Medicine, Sri Manakula Vinayagar Medical College and Hospital, Puducherry, India, **2** Department of Community Medicine, Yenepoya Medical College, Yenepoya (Deemed to be University), Mangaluru, Karnataka, India, **3** Department of Community and Family Medicine, All India Institute of Medical Sciences, Bathinda, Punjab, India, **4** Department of Respiratory Medicine, Madras Medical College, Chennai, Tamil Nadu, India, **5** Division of Health Systems Research, ICMR-National Institute of Epidemiology (ICMR-NIE), Chennai, Tamil Nadu, India, **6** State TB Cell, Government of Puducherry, Puducherry, India

* grevadi07@gmail.com

## Abstract

*Ni-kshay Mitras* are volunteers who provide monthly food baskets worth 7.5-10 USD to people with tuberculosis as a part of India's recent community-driven nutritional support initiative. We assessed the initiative's coverage and implementation challenges in Puducherry, India. This mixed-methods study involved secondary data of adults with TB notified from public facilities between April 2023 and March 2024. This was followed by 42 in-depth interviews (qualitative) with the stakeholders from January to February 2025. Data analysis was done using the R software. Manual thematic analysis was done to derive qualitative insights using Kurt Lewin's framework. A total of 131 *Ni-kshay Mitras* were registered. Of the 1055 adults with TB who consented to receive nutritional support, 517(49%) received at least three food baskets during their treatment period, similar across different socioeconomic and nutritional statuses. Duration between the diagnosis and receipt of the first food basket (N = 852) was 41(IQR 26,71) days. Facilitators, barriers, and suggested solutions were grouped under major categories such as donor-related, health system-related, and those related to the perception of patients with TB. Facilitators included an established protocol for distributing food baskets, home delivery for sick/older adults, quality check of food baskets, and patient feedback receptiveness. Donor-related challenges included a lack of consistency, reminder requisition, and an inclination for distribution within their geographical settings. Health provider-related challenges were difficulties in monthly collection and transportation of the supplies, hesitancy to approach the donors, poor peer support, and challenges in data documentation on the reporting

**Data availability statement:** The minimal data set underlying the findings of this study contains sensitive and identifiable information about persons with tuberculosis and cannot be shared publicly to protect participant confidentiality. Data access is therefore restricted by the State Tuberculosis Office, Government of Puducherry. Researchers who meet the criteria for access to confidential data may submit a reasonable request in writing to the State Tuberculosis Officer (Email: stopd@rntcp. org) through the Department of Community Medicine, Sri Manakula Vinayagar Medical College and Hospital, Puducherry (Email: info. commedicine@smvmch.ac.in). The corresponding author will facilitate communication but will not be responsible for approving access.

**Funding:** The modular training under this SORT IT course was funded by the ICMR-National Institute of Epidemiology (ICMR-NIE), Chennai, India. ICMR-NIE did not provide any specific funding for the operational/implementation research (that resulted in this manuscript) conducted through this SORT IT course. The research was conducted in routine operational settings utilizing existing health system resources and workforce. Open access publication fee was supported by ICMR-NIE Chennai, India.

**Competing interests:** The authors have declared that no competing interests exist.

portal. Nearly half of the adults with TB received at least three food baskets over their treatment period, with prolonged latency in initiating the first food basket. Gaps were observed in prioritizing food basket distribution to the vulnerable group of patients with TB.

## Introduction

In India, undernutrition and tuberculosis (TB) share a bidirectional relationship where 34% of all incident TB is attributed to undernutrition [1]. The increased demand for energy in people with active TB further leads to loss of weight and predisposes them to delayed sputum conversion, malabsorption, drug toxicity, relapse, and mortality [2,3]. The exacerbated nutritional demand in an earning member with TB can further perpetuate financial burden, food insecurity and poverty in the family [4]. Various strategies to alleviate out-of-pocket expenditure and provide nutritional supporting TB-affected households, like cash transfers, additional rations and ready-to-use therapeutic foods for patients and household contacts, have been studied [5–11]. National Tuberculosis Elimination Programme (NTEP) program in India launched the *'Nikshay Poshan Yojana'* initiative in 2018 to support the nutrition needs of patients with TB (PwTB). Under this initiative, PwTB receive 1000 INR (~ 12 USD) per month through direct benefit transfer during their course of treatment [12]. Within this endeavour, *'Ni-kshay Mitra*s' (friends for ending TB) initiative was launched under *'Pradhan Mantri TB Mukt Bharat Abhiyaan'* (Prime Minister's TB free campaign) in September 2022. *'Ni-kshay Mitra'* refers to community-based individuals, non-governmental organisations (NGOs),cooperative societies, and elected representatives who volunteer to support PwTB under this initiative. They also motivate the patients to adhere to the treatment and improve treatment outcomes [13].

Challenges and treatment outcomes underlying this initiative have been studied within specific geographical settings in both global and Indian context [14–16]. Evidences suggest that there is differential preference and irregularity in the supply chain of the food baskets from the donor's side and acceptability from adults with TB, resulting in sub-optimal utilisation [10]. There is limited information about the total food baskets received by PwTB during their entire course of treatment. There is also a research gap in determining the time interval between treatment initiation and the distribution of food baskets, as the majority of TB deaths occur within the first two months of treatment [17]. This being a recent initiative, it is yet to be explored widely in Puducherry district (population ≈1.3 million)in southern India following the implementation in 2022. More studies are required from various geographies in India to guide policy at the national and state levels. Our study adds to this evidence base in routine health system settings.

From a multistakeholder perspective, we assessed the coverage and implementation challenges of the community-driven *Ni-kshay Mitra* initiative in delivering nutritional supplements to adults with TB in Puducherry, India. The objectives were to i) describe the characteristics of *Ni-kshay Mitras* and document the various models of

food basket distribution, ii) determine the coverage of food basket distribution during the first 6 months of treatment and time interval in the initiation of first food basket from diagnosis, and iii) understand the facilitators and barriers to implementation from the perspective of health care providers, *Ni-kshay Mitras* and adults with TB.

## Materials and methods

### Ethics statement

The institutional ethics approval was obtained from ICMR National Institute of Epidemiology, India (NIE/IHEC/A/202408-08 dated 18/9/2024) and Sri Manakula Vinayagar Medical College and Hospital, Puducherry (SMVMCH-ECO/AL/385/2024 dated 24/10/24). A waiver of written informed consent was sought for secondary data. The ethics committee had approved the written informed and telephonic/distant informed oral consent process for qualitative interviews. The latter was documented in the informed consent form by the investigators, SM and SWP, in the presence of PI RG. State (Puducherry UT) operational research committee approval (dated 24/9/2024) and administrative approval from the state TB office were obtained to access secondary data (PSHS/NTEP/PA/2024-25/ dated 1/10/2024).

### Study design

This mixed methods study included secondary data in the quantitative phase, followed by in-depth interviews in the qualitative phase (sequential explanatory) [18].

The secondary data included information on a cohort of adults with TB notified at public facilities between 1st April 2023 and 31st March 2024. The exposure of interest was whether individuals had consented to or declined nutritional support. Their follow-up outcomes included information about the number of food baskets received throughout their treatment period, treatment outcomes, and changes in weight gain. Following secondary data analysis, in-depth interviews with the stakeholders were conducted between 1st January 2025 and 25th February 2025 (Fig A in S1 Text).

### Study settings

The study was conducted in Puducherry district with a population density of 3232 per square kilometer and literacy rate of 85.4% [19]. The presumptive TB examination rate was 19530 per million population, and the TB case notification rate was 2635 per million population for the year 2024, compared to 17100 per million and 1788 per million population at the national level [20].

Since September 2022, the peripheral health institutions (PHIs, n = 34) comprising 32 primary health centres and two community health centres have been providing nutritional support in addition to TB treatment. Tertiary care health facilities for diagnostic and treatment support include a district hospital, a TB chest clinic, a government hospital for chest diseases, and eight medical colleges (two government and six private). The district has four TB units (TU, a sub-district administrative unit for TB program implementation).

Those notified adults with TB undergoing treatment were approached by the healthcare providers for opting for nutritional support using a consent form [4]. The consent was obtained either physically by signature/ thumb impression from adults with TB or through sharing of the one-time password generated through the *Ni-kshay* portal (web-based surveillance platform for TB in India) to the healthcare providers (OTP based consent). *Ni-kshay Mitras* were registered through the *Ni-kshay Mitra* module within the *Ni-Kshay* portal with the affirmation that the details of the beneficiaries shared with them shall not be used for any other purpose or shared with any organisations/individuals other than the NTEP [21]. In this way, *Ni-kshay Mitras* were linked to the consented adults with TB within their geographical setting.

Each *Ni-kshay Mitra* can adopt a minimum of one consented patient on treatment for at least six months. During this period, they are expected to distribute locally available and nutrient-dense food baskets to the patient on a monthly basis, which usually costs between INR 600–800 (7.5-10 USD) per basket [4]. The basket is expected to have at least 3 kg of cereals and millet, 1.5 kg of pulses, vegetable oil of 250 ml, groundnuts or milk powder of 1 kg, vegetables or fruits,

vitamin B and mineral tablets [4].The various models of existing food basket distribution in India has been given provided in Fig B in S1 Text [13].

## Study population

**Quantitative phase.** The quantitative phase included secondary data of all adults with TB notified in public health facilities (n = 34). The time period from April 2023 was chosen as the initiative was launched nationally six months earlier in September 2022. We intended to follow up with all individuals who consented to receive nutritional support till their treatment completion, which varied between six months and one year. Additionally, basic information about all *Ni-kshay Mitras* providing nutritional support between April 2023 and October 2024 was included.

**Qualitative phase.** Senior Treatment supervisors (STS, stationed at TB unit level) and TB health visitors (TBHV, at PHIs with high notifications, to support existing STS) from the TB program who were directly engaged in food distribution, stock maintenance, and reporting in *the Ni-Kshay Mitra* module within the *Ni-Kshay* portal were included. The STS in charge of each TB unit was included (total four), whereas two TBHVs under each STS were purposively selected through maximum variation sampling, amounting to a total of eight people [22].Three Medical Officers selected through extreme or deviant sampling from PHIs had participated [22].To explore the variations in food basket coverage, we intended to generate insights from those PHIs with high and low coverage of food baskets distribution. Four individual *Ni-kshay Mitras* were purposively selected through extreme or deviant sampling till data saturation was achieved. One elected representative and one representative from an institution, two representatives each of NGOs, corporates, and community were included [22].Ten adults with TB, a combination of those who consented and five adults with TB who declined nutritional support, were included through convenience sampling. Hence, a total of 42 interviews were conducted, comprising of 33 face-to-face and 9 telephonic interviews with various stakeholders, including TB program personnel.

## Data collection

**Quantitative phase.** Secondary data from TB treatment cards archived in the State TB cell were single-entered in the mobile-based EpiCollect5 application. Undernutrition was defined as body mass index (BMI)<18.5 kg/m$^2$ and severe undernutrition as body mass index <16 kg/m$^2$. TB treatment cards also had information on the socio-economic status of the adults with TB classified as above or below the poverty line based on the Government of India norms. Secondary data related to the number of food baskets from the *Ni-kshay Mitra* register were single-entered in MS Excel. Data related to the *Ni-kshay Mitras* and their consent for nutritional support was extracted from the *Ni-kshay Mitra* module within the *Ni-kshay* portal. *Ni-kshay's* unique ID of the patient was used to trace information from various sources and to compile information in the Excel sheet. Confidentiality was maintained by storing the electronic data in a password-protected computer accessible only to the investigators. Data quality was assured by checking for duplicate entries and for completeness and accuracy through verification from other sources. Those data that align with the study period were sourced from a routinely updated treatment card, register, and portal for timeliness and relevance [23].The available case analysis approach was used to handle the missing data.

**Qualitative phase.** After piloting, the interview guides in the vernacular (Tamil) language (S2 Text) were validated for their content with three subject experts prior to the interview. We conducted one-to-one in-depth interviews with STS, TBHV, Medical Officers, *Ni-kshay Mitras,* and adults with TB. These were conducted in person at the PHIs and over the phone, as we assumed the quality of interviews to be the same irrespective of the place of interview due to operational constraints. The audio-recorded responses were stored anonymously with maximum confidentiality by the PI. There were no repeat interviews conducted with the stakeholders. Adults with TB were interviewed at PHIs during their routine expected visit. The interview guide was shared in advance, and the interviewees were given an option to choose the mode of interview as per their availability on a scheduled date and time.

The interviewer(s) formally introduced themselves, followed by a detailed explanation of the study's purpose. Three interviewers (RG, SM, SWP) were trained in qualitative research. They conducted the interviews and audio-recorded the information. The interviewer took notes simultaneously during the meeting and summarized them to the interviewee before closing the session. Each interview lasted 20–30 minutes.

### Data analysis

**Quantitative phase.** All data sets in Microsoft Excel were merged using the unique *Ni-kshay* identifier and analyzed using R software (v4.4.2; R Core Team 2021) [24]. The continuous variables were summarised as median (interquartile range) and the categorical variables as number (%). Frequency and proportion were used to depict the care cascade: how many consented, how many were provided with food baskets, and for how many months. Median and IQR were used to summarize the monthly food baskets per adult with TB and the interval between treatment initiation and receipt of the first basket. These indicators were presented by socioeconomic and nutritional status. Coverage was defined as the number of PwTB who have received at least one food basket among all those who have consented to the nutritional support. Equity was assumed if these indicators were better among adults with very severe undernutrition and below the poverty line. Negative binomial mixed model regression was used to identify the predictors (socio-demographic, clinical, and diagnostic characteristics – fixed effect) of receiving less than the median number of food baskets (skewed distribution) after accounting for the random effect of 34 PHIs. Variables with a p-value less than 0.20 and those variables that were the basis of prioritisation of food basket distribution were applied for adjusted analysis.

**Qualitative phase.** Qualitative data was reviewed by reading the notes and actively listening to the interview recordings (Fig 1). All the audio-recorded interviews and the handwritten field notes were transcribed and translated from vernacular language to English manually within a week's interval by researchers RG, SM, and SWP. The qualitative data

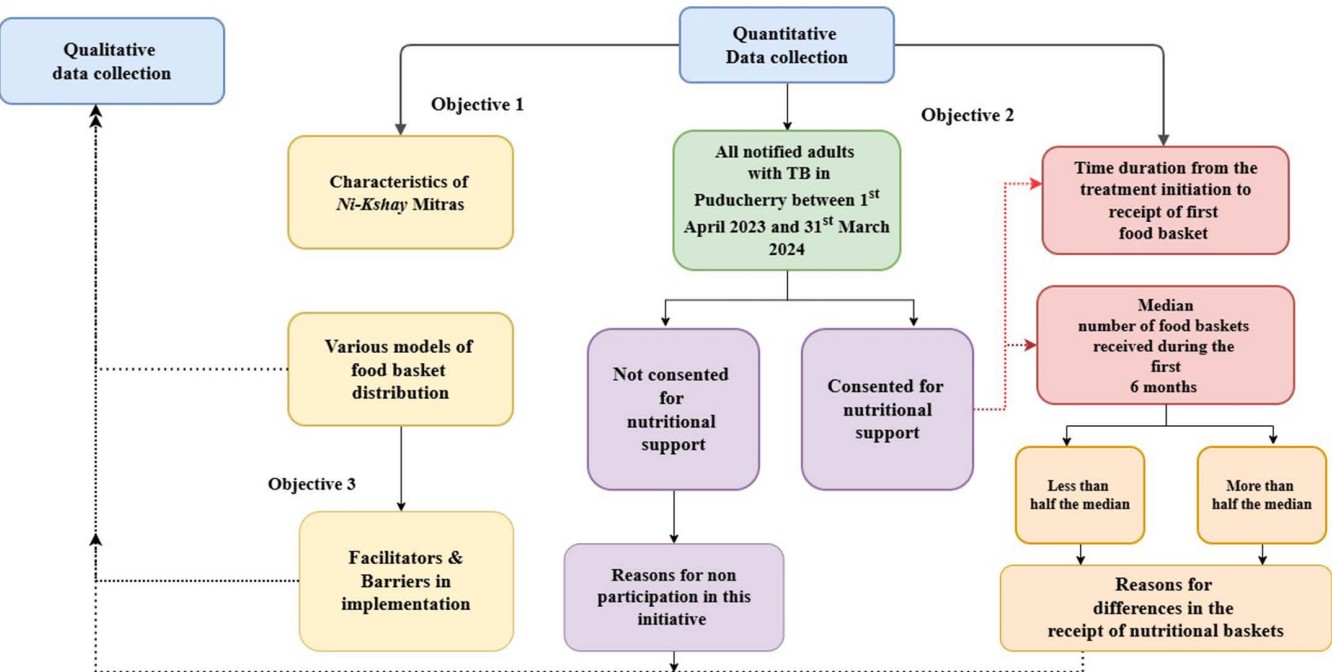

**Fig 1. Integration of mixed methods study design to address the coverage and implementation challenges of the nutritional support initiative to adults with TB in Puducherry district, India, between October 2024 and February 2025.**

were organized into easily retrievable sections as per topic guides. Descriptive codes were identified by researchers SM and SWP from the transcripts using the free and open-source Taguette software [25].This software facilitated comprehensive documentation of the findings from various stakeholders and enabled verification whenever necessary. Researcher RG reviewed the codes (codebook provided as an S1 Codebook), and manual content analysis was done along with researcher AD, with expertise in qualitative analysis. The Kurt Lewin's framework of force field analysis was used to analyse the driving forces or facilitators, and restraining forces or related to distribution of food baskets and solutions for its improvement [26].The codes from the qualitative interviews (S1 Codebook) led to the categories identified broadly as: donor-related barriers, health system-related barriers, perception outcomes and donor expectations which was further classified into sub categories [27]. The first author (RG) assessed saturation during analysis, with meaning saturation deemed reached when no new nuances or dimensions emerged from the data [28].The results were documented and reported as per the GRAMMS checklist (S1 GRAMMS Checklist) [29].

## Operational definitions (S1 Text)

### Results

**Characteristics of *Ni-kshay Mitras*.** A total of 162 *Ni-kshay Mitras* provided nutritional support. From the available details of 131 *Ni-kshay Mitras*, 101(77.1%) were individuals, 10 (7.6%) were institution-based, 8 (6.1%) NGOs, 4 (3.1%) corporates, 3(2.3%) elected representatives and 5(3.8%) from companies, societies and clubs. Six individuals, one institution and one NGO had opted for a minimum duration of 12 months and the rest for 6 months for the distribution of food baskets. This information served as a basis for purposively selecting the Ni-Kshay Mitra's for in-depth interviews.

**Models of food basket delivery.** Fig 2 describes the 5 food basket delivery models in Puducherry, based on interviews with TB program personnel (TBHVs and STS) and *Ni-kshay* Mitras: model 1 by the NGOs, model 2 by the corporate group, model 3 by *Ni-kshay Mitras* from health care settings, model 4 by the institutions and community *Mitras*, and model 5 by the others. At the level of PHIs or the state TB office, the respective TB program personnel handled the baskets in the PHIs. *Ni-kshay Mitras* provided the food baskets to the PHIs (directly or via the state TB office). In some instances, the *Ni-kshay Mitras* directly supplied the food baskets to the adults with TB within the PHIs. This was also documented in *the Ni-Kshay Mitra* module within *the Ni-kshay* portal and the *Ni-Kshay Mitra register*.

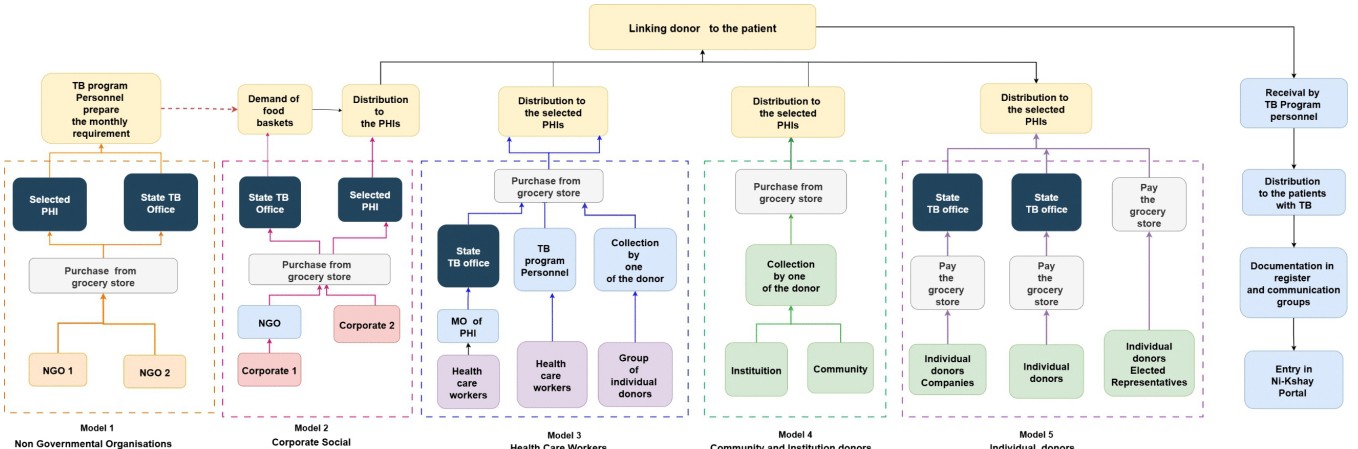

**Fig 2. Various models of food basket distribution to the adults with TB in Puducherry district, India, between April 2023 and October 2024.** NGO – Non-Governmental Organization, PHI – Peripheral Health Institutions, MO – Medical Officer In charge, TB program personnel – (TBHVs, STS).

**Coverage and time interval in the initiation of the first food basket.** Of the1283 adults notified with TB, 1116 (86.9%) were from PHIs. Of the 1088 available entries in *Ni-Kshay*, 1055 (97%) had consented to nutritional support (Fig 3). Of 1055, the median age was 47 years (IQR 37,59), and 671 (63.6%) were males. A total of 739 (70%) were below the poverty line, 223 (21.1%) and 183 (17.3%) had undernutrition and severe undernutrition, respectively (Table 1). Among those with complete data available on socioeconomic status and BMI (N = 152), 91.4% of adults with severe undernutrition belonged to the below poverty line group. Of the 1055 individuals who consented,517(49%) received at least three food baskets, and 70(6.6%) received six or more. We did not look for the association between treatment outcomes and the receipt of food baskets in our study. A total of 852 received the first food basket after a median of 41(IQR 26,71) days from diagnosis. Negative binomial regression did not identify any significant predictors of receiving fewer than or equal to two food baskets (Table A in S3 Text and Table B in S3 Text).

The following verbatims highlight the insufficient number of donors and its effect on the distribution of food baskets from the health care provider's perspective.

*"This initiative is completely dependent on the donors. If the donors provide monthly, then the distribution of food baskets will increase. There are more companies in the area surrounding our health Centre. They said they will ask their seniors and let us know*. But currently, no one came forward" (TBHV 3).

*"The difference in the distribution of food baskets within the health facilities of my area was largely due to a smaller number of donors*" (STS 4).

Qualitative findings (Table 2) reveal that some of those who did not provide consent appeared to be from better socio-economic backgrounds. Then, there were technical issues in getting the one-time password (OTP) from the patient (for consent). Some of them were not aware of the monthly provision of food baskets.

*The nurse also informed us about the provision of food baskets to adults with TB, like me, but I refused it. I am from a middle-class family, so we have no issues. They can give it to those who really need it; what will we do if we get it?"* (Adult with TB, 35–45 years).

*"Some difficulties persist in getting consent, such as many are not advanced enough to receive the OTP. So, when we asked them to tell us the OTP they had received on their phone, they couldn't do that"* (TBHV 6, female).

**Ensuring the quality standards of the contents within the food basket.** The *Ni-kshay Mitras* purchased items with certified trademarks and were also receptive to the feedback provided by PwTB.

*"I want to ensure the quality, so I will give the TB program personnel my contact card and let them know that they can call me anytime or WhatsApp me, so whatever complaints they have, received from the adults with TB they can let me know, and we will rectify it as soon as possible"* (NGO Ni-kshay Mitra).

*"They had the **XXX trademark,** madam, while I purchased the materials. Hence, I bought them and gave them to the health care workers"* (Individual NM 1).

**Feedback about food baskets from the consented patients with TB.** Feedback from PwTB about the quality of food baskets received from Ni-kshay mitras was positive. However, they believed that the quantity was dependent on the number of family members consuming it.

*"I have made a good use of it; the items were of a higher quality than those provided in ration (public distribution) shops. I have made Porridge, Upma, boiled chickpea dishes, and it was useful"* (Adults with TB, 55-65 years).

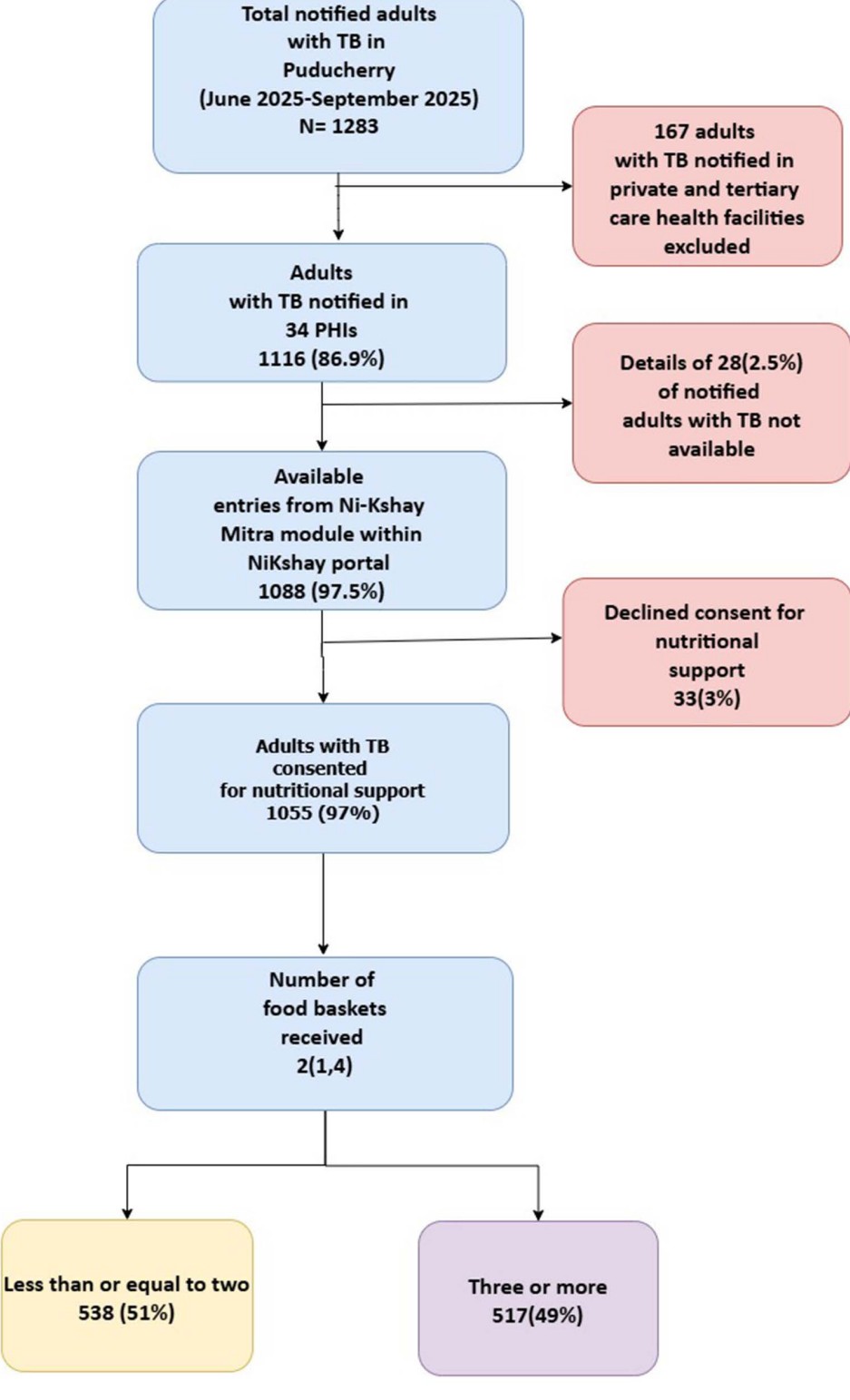

**Fig 3. Implementation cascade of nutritional support to the adults with TB in Puducherry district, India, between April 2023 and October 2024.**
TB- tuberculosis, PHI – Peripheral Health institutions.

**Table 1. Socio demographic and clinical characteristics of the adults with TB consented for nutritional support between April 2023 and March 2024, Puducherry district, India (N = 1055).**

| Characteristic | n | % |
|---|---|---|
| **Individual characteristics** | | |
| Age (years) in median (IQR) | 47(36,59) | |
| Gender | | |
| Males | 671 | 63.6 |
| Females | 380 | 36 |
| Transgender | 4 | 0.4 |
| Residence | | |
| Rural | 323 | 30.6 |
| Urban | 597 | 56.6 |
| Not recorded | 135 | 12.8 |
| Occupation | | |
| Working | 481 | 45.6 |
| Not working | 193 | 18.3 |
| Not recorded | 381 | 36.1 |
| **Family level characteristics** | | |
| Socio Economic status | | |
| Above poverty line | 121 | 11.5 |
| Below poverty line | 739 | 70 |
| Not recorded | 195 | 18.5 |
| No of Household contacts in median (IQR)[a] | 3(1,3) | |
| **Behavioural characteristics** | | |
| History of Tobacco use | | |
| No | 893 | 84.6 |
| Yes | 135 | 12.8 |
| Unknown | 27 | 2.6 |
| History of alcohol use | | |
| No | 828 | 78.5 |
| Yes | 192 | 18.2 |
| Unknown | 35 | 3.3 |
| BMI at baseline | | |
| Beyond 18.5 | 643 | 60.9 |
| Undernutrition (16-18.4) | 223 | 21.1 |
| Severe undernutrition (<16) | 183 | 17.3 |
| Not recorded | 6 | 0.6 |
| **Diagnosis and Treatment details** | | |
| Site of Tuberculosis | | |
| Pulmonary | 704 | 66.7 |
| Extra pulmonary | 347 | 32.9 |
| Both | 4 | 0.4 |
| Type of Tuberculosis | | |
| New | 905 | 85.8 |
| Retreatment | 132 | 12.5 |
| Drug resistant | 18 | 1.7 |

*(Continued)*

**Table 1.** (Continued)

| Characteristic | n | % |
|---|---|---|
| Classification of Tuberculosis | | |
| Clinically diagnosed | 289 | 27.4 |
| Bacteriologically confirmed | 723 | 68.5 |
| Not recorded | 43 | 4.1 |
| Drug combinations taken | | |
| Fixed dose combinations | 98 | 193 |
| Loose drugs | 69 | 6.5 |
| Not recorded | 5 | 0.5 |
| **Comorbidity status** | | |
| HIV status | | |
| Non-reactive | 1,031 | 97.7 |
| Reactive | 3 | 0.3 |
| Unknown | 8 | 0.8 |
| Not recorded | 13 | 1.2 |
| Diabetes Mellitus | | |
| No | 468 | 44.4 |
| Yes | 389 | 36.9 |
| Unknown | 180 | 17 |
| Not recorded | 18 | 1.7 |

[a]Data not available for 26 adults with TB.

"We are a 3-member household, and the quantity is sufficient for us. For households with 4 or 5 members, especially if they're sharing with others, it might not be enough. In our case, the supply lasts around 20 to 25 days, as we receive it for 2 people." *(Adults with TB, 60-70 years)*

Some *Ni-kshay Mitras* took steps to ensure a monthly supply of food baskets, and nearly half of the adults with TB received at least three food baskets.

"*We will fix a date that is generally the 25th of every month by when the order should have been placed to the shop; then, during the first of every month, we will get those baskets from the store and subsequently coordinate for distribution" (Individual Ni-kshay Mitra, HCP 2).*

Healthcare providers stated stigma as one of the reasons for the adults with TB declining and also not turning up to receive food baskets in subsequent visits. Some donors lose enthusiasm over time.

"*People don't want others to know that they are taking tablets for TB. So, if they come here, and get provisions during the program we are conducting, then others would get to know about their illness"* (STS1, 38-48 years).

"*The donors start with enthusiasm, but after time goes on, they lose the enthusiasm"* (TBHV6, female).

The reason for the wider time interval between the diagnosis and receipt of the first food basket, as explained by the health care provider, was:

**Table 2. Perceived facilitators, barriers and suggested solutions in providing nutritional support to adults with TB from stakeholder's perspective in Puducherry district, India during January-February 2025.**

| Sub categories | Perceived barriers | Facilitators | Suggested solutions |
|---|---|---|---|
| **Donor related** | | | |
| Problems with donor identification | Less number of donors (*STS 1,3 and Medical Officer*)<br>Preference to donate in the respective area (*TBHV 5*)<br>Difficulty in contacting donors directly by the HCPs (*TBHV 3*) | Approaching with individuals who have rapport in the community. (*STS 2*)<br>Approaching during campaigns. (*Medical officer 3, TBHV 3*) | Approaching factories, elected representatives, NGOs, Government staffs and HCWs. (*TBHV 1,3; MO 1,3,4 and STS1,3*)<br>Engaging medical officers in identifying donors. (*MO 4, STS 1–3*)<br>Promotion through IEC and by health talks for increasing public awareness. (*TBHV 1, MO 4, HCP 1, Individual NM 3*)<br>Promotion through social media handles |
| Expectations from the donors | Doubts about food baskets reaching the patients (*TBHV 5*)<br>Preference to donate in the respective area (*TBHV 5*)<br>Donors asking for patient details<br>Donors requesting reminders (*NGO 2 Ni-Kshay Mitra*) | Established protocol for food distribution in Puducherry (*TBHV 5*). | |
| Shortage of supply | Delay from donors (*STS 3*)<br>Lack of consistency (*TBHV 1*) | Coordination with other PHCs to ensure supply (*TBHV 1*)<br>Mutual cooperation among the adults with TB. (*TBHV 8*) | Stakeholders meeting and allocating budget in advance. (*Corporate 1 and 2, NGO 2*)<br>Monthly provision of nutritional kit by government. (*TBHV 1 and 3, STS 3, community mitra 1, NGO 2*) |
| Sustaining the donor engagement | Fading enthusiasm among the donors (*TBHV 6*)<br>Hesitancy among female HCPs to repeatedly approach the donors. (*Individual NM, HCP 1*) | Inviting the donor for the distribution of food supplements.(*STS 1 and TBHV 6*).<br>Sharing reports of number of food basket distribution. (*Corporate 1 Ni-Kshay Mitra*)<br>Constructive feedback to the donors<br>Recognition of donors by various program managers (*STS 1*). | Monitoring the weight of adults with TB and reporting their treatment outcomes. (*NGO 2, individual mitra 1, MO 6*). |
| **Health system related** | | | |
| Transport and delivery challenges | Non-availability of transport from the State TB cell to the respective PHIs in some instances. (*TBHV 1, STS 1*).<br>No proper storage facility at PHIs (*TBHV 1*). | Home delivery for sick/old people and those living at distant places (*TBHV 9*). | |
| Feeling of Burn out by the HCWs | Difficulties in monthly collection of food baskets.<br>HCWs losing motivation due to repeated persuasion. (*TBHV 8*)<br>Poor support from the peers (*STS 3*) | | Establishing a system at the State TB cell to receive payments which could be dedicated towards purchase of food baskets. (*TBHV 3 and NM HCP 2*) |
| Documentation and reporting issues | Entries in portal following distribution of food basket is time consuming. (*TBHV 2*)<br>Issues related to receipt of validation of one-time password by adults with TB. (*STS 1*) | Recording in the *Ni-Kshay Mitra* register first followed by entry in the portal (*TBHV 3*). | |

*(Continued)*

**Table 2.** (Continued)

| Sub categories | Perceived barriers | Facilitators | Suggested solutions |
|---|---|---|---|
| **Related to perception of adults with TB** | | | |
| Related to the contents of food basket | Sense of insufficient food items. (*Community NM, TBHV 1 and 3*) | Flexibility to change, based on needs. (*Medical Officer 1*) Receptive to the feedback from adults with TB. (*STS 3*) Providing checklist of food items (*Institution Mitra*). Checking the food items for the quantity and quality during purchase by the donors (*Ni-Kshay Mitra HCP 3, TBHV 6, STS 1 and 2*) Inspecting the food baskets at PHIs by HCPs. (*STS 1*) | Adding variety in the food baskets. (*TBHV 8, MO 2, individual NM 2*) Increasing the quantity of the items within the food baskets. (*TBHV 4, MO 5, community NM 2, adults with TB 4*) |
| Hesitancy to receive from public facility | Hesitancy by the adults with TB to receive food baskets from PHIs. (*TBHV 5 and 7*) | Ensuring privacy of adults with TB. (*TBHV 5*) Patient bystanders collecting food baskets (*TBHV 1 and 3*). | Distribution in the presence of medical officers. (*TBHV 6, MO 4, individual Ni-Kshay Mitra 3*) Providing food baskets alongside monthly medicines. (*TBHV 6, NM HCP 1, MO 6, adult with TB 9*) Involving patient bystanders for supportive care. (*Adult with TB 6, NGO 2, MO 6*) Extending nutritional support to bystanders. (*Individual NM 2*) |
| Expectations by adults with TB | Expectation about issue of next food basket by adults with TB (*TBHV 2*). | Transparency in distribution of food baskets. (*STS 4*) Attempting for timely provision of food baskets. (*STS 3*) | Prioritising food baskets to the vulnerable. (*TBHV 5, MO 4, STS 2, community mitra 1*) Increasing the number of food baskets provided. (*TBHV 8, STS 1*) |

*"Time to initiation of food baskets is highly based on the donors. We generally issue food baskets only once a month. Otherwise, we will schedule it for next month." (STS 3, 38-48 years).*

To address the monthly interrupted supply, some staff suggested increasing the quantity in the food baskets and acknowledging the need for more *Ni-kshay Mitras* to sustain nutritional support.

*"Quantity can be increased, like 5 kg of rice in place of 3 kg, and also increase the number of millets and proteins like Dal. This is because it will be used by the family of those adults with TB during interrupted supply" (TBHV 5, female).*

*"The government should also acknowledge more Mitras with a trophy, as a few people consider it a pride to keep it at home. It was not my expectation initially when I had enrolled in this initiative, but they later gave, which was actually good" (Individual Ni-Kshay Mitra, HCP 3).*

**Equity in the provision of food baskets.** Food basket provision stratified by nutritional and socio-economic status is depicted in Table 3. Marginal prioritization (P 0.091) of adults with TB belonging to below the poverty line was observed with respect to food basket provision. However, this was not significant after adjusting for potential confounders (age, residence, history of tobacco use, status of diabetes, nutritional status, site and type of tuberculosis, and its classification). The time interval between the diagnosis and receipt of the first food basket was lesser for vulnerable groups, i.e., severe undernutrition and below poverty line adults.

Qualitative data suggested that some prioritization happened based on socioeconomic status (Table 3).

PLOS Global Public Health

**Table 3. Distribution of food baskets to adults with TB notified from peripheral health institutions during April 2023-October 2024 in Puducherry district, India, by nutritional status and socio-economic statuses.**

| Indicator | | | | |
|---|---|---|---|---|
| **Adults with TB who have received monthly food baskets at least one [N=1055]** | N | n | (%) | P value[a] |
| Nutritional status[c] | | | | |
| Overall | **1049** | 848 | (80.8) | 0.3 |
| Normal or overweight | 643 | 508 | (79) | |
| Undernutrition (not severe) | 223 | 188 | (84.3) | |
| Severe undernutrition | 183 | 152 | (83.1) | |
| Socio economic status[d] | | | | |
| Overall | **860** | 728 | (84.7) | 0.2 |
| Above poverty line | 121 | 99 | (81.8) | |
| Below poverty line | 739 | 629 | (85.1) | |
| **Number of food baskets received by adults with TB [N=1055]** | N | Median | (IQR) | P value[b] |
| Nutritional status[c] | | | | |
| Overall | **1049** | 2 | (1,4) | 0.8 |
| Normal or overweight | 643 | 2 | (2,4) | |
| Undernutrition (not severe) | 223 | 2 | (1,4) | |
| Severe undernutrition | 183 | 2 | (1,4) | |
| Socio economic status[d] | | | | |
| Overall | **860** | 3 | (1,4) | 0.091 |
| Above poverty line | 121 | 2 | (1,4) | |
| Below poverty line | 739 | 3 | (1,4) | |
| **Number of food baskets received by adults with TB [N=852]** | | | | |
| Nutritional status[e] | | | | |
| Overall | **848** | 3 | (2,4) | 0.4 |
| Normal or overweight | 508 | 3 | (2,4) | |
| Undernutrition (not severe) | 188 | 3 | (2,4) | |
| Severe undernutrition | 152 | 3 | (2,4) | |
| Socio economic status[f] | | | | |
| Overall | **728** | 3 | (2,4) | 0.2 |
| Above poverty line | 99 | 3 | (2,4) | |
| Below poverty line | 629 | 3 | (2,4) | |
| **Time from diagnosis to the receipt of first food basket (days) [N=852]** | N | Median (IQR) | (IQR) | P value[b] |
| Nutritional status[e] | | | | |
| Overall | **848** | 41 | (26,71) | 0.5 |
| Normal or overweight | 508 | 41 | (26,69) | |
| Undernutrition (not severe) | 188 | 44 | (28,73) | |
| Severe undernutrition | 152 | 39 | (23,80) | |
| Socio economic status[f] | | | | |
| Overall | **728** | 41 | (25,68) | >0.9 |
| Above poverty line | 99 | 43 | (26,67) | |
| Below poverty line | 629 | 40 | (25,69) | |

[a]Chi squared test,

[b]Kruskal Wallis rank sum test,

[c]6 data not available,

[d]195 data not available,

[e]4 data not available,

[f]124 data not available.

*"We prioritise the people living below the poverty line, like the people with single-earning members, etc. So, except for the people who work in the office and the well-to-do, we distribute the provisions we receive for the rest of them"* (TBHV 2, female).

*"We generally give to adults with TB who are poor or weakly built because of limited supplies. Those who were coming from far areas where they have difficulty in travelling, we distribute them at their homes depending on the availability of transport"* (STS 3,38-48).

During the shortage, it appeared that adults with microbiologically confirmed pulmonary TB were prioritized.

*"In situations of shortage, we tend to give to the positive adults with TB and not extrapulmonary cases because in extrapulmonary cases, it won't be spreading to others, right? But in pulmonary, if the person is deprived of nutrition, then he/she tend to spread more… also they aren't able to go to work due to the nature of the disease"* (TBHV 6, female).

**Weight gain by the median number of food baskets received.** Of 852 who received at least one food basket, weight was documented at the end of the intensive phase in 612 (71.8%) and at the end of treatment in 184 (21.6%). Hence, we were not able to document weight gain stratified by receipt of food baskets and attribute any changes in outcomes to food baskets. The challenges related to repeated weight measurements during the follow-up were attributed to absenteeism by the adults with TB while receiving medications and food baskets.

*"Mostly, we'll do it. Initially, they'll come till the end of the intensive phase. After some time, they'll feel better, so they leave for work. In that case, the wife or brother will come and get the tablets"* (TBHV 3, female).

Not having reliable information from the program regarding weight gain and treatment outcomes might have dissuaded some Ni-kshay Mitra from continuing their support.

*"Budget planning starts at the beginning of the year or at the end of the year. So, we need the list of the adults with TB whom we have provided nutritional support to, and the improvements seen in them. So that we can get help from the corporate side and they can allot some more budget to this program"* (Representative, Corporate Ni-kshay Mitra).

## Discussion

Our study is amongst the few studies from India that have systematically assessed the community-driven *Ni-Kshay Mitra* initiative's coverage and implementation challenges, and we present certain interesting findings. *First*, about eight of every ten adults with TB consented to nutritional support services. *Second,* nearly half of the consented adults with TB received at least three food baskets during their treatment period, with a broader interval in the service initiation. *Third*, we observed gaps in prioritization attempts to ensure equitable distribution of food baskets to the vulnerable groups.

### Cascade of food basket distribution and delivery care model

We observed a higher proportion of adults with TB who had consented and subsequently received nutritional support compared to a previous Indian study [15].Innovative approaches like flexible food basket procurement and delivery models in Puducherry can explain the better coordination with various donors. Such models stand out from those

implemented in other states of India. In Gujarat, an agency/third party was engaged in delivering food baskets, while in Delhi, a dedicated web platform was operationalized to facilitate donations [13].Globally, similar community-based food assistance programme was implemented in Afghanistan through the support from NGOs and the elderly within the community [30].

### Addressing equity in distribution

Adults with TB who received two or fewer food baskets had the first basket provided after almost one and a half months following diagnosis due to variations in the supply chain across the TB units. In such a scenario, we expect that the adults with severe undernutrition should be prioritized and supplied with food baskets throughout their treatment period [2]. However, we observed that all adults with TB were distributed food baskets irrespective of their nutritional and socio-economic status since 2022. Additionally, the majority of the adults with severe undernutrition lived below the poverty line, necessitating the need of targeted criteria and real-time dashboards to ensure equity. Evidence from the RATIONS trial suggests that a 5% increase in weight at the end of two months in PwTB on nutritional support is associated with reduced hazard of death [31]. Thus, early initiation of nutritional support in PwTB improves the nutritional status and treatment success [31,32].

### Monitoring weight gain among those who received food baskets

We remain inconclusive about changes in the weight of adults with TB during the course of the treatment and the absolute impact of nutritional support due to limited data availability. Documenting weight at baseline and changes at monthly follow-up provides insights about weight gain following nutritional support [31,33]. Despite the availability of logistics and the provision for documentation of weight in the treatment card, a documentation gap exists. The reason is absenteeism by PwTB during follow-up visits and the receipt of food baskets by their family members. Counselling PwTB on the importance of weight measurement during follow-up visits, along with real-time documentation by health care providers, is crucial for bridging the gap.However, the impact on treatment outcomes was not observed, unlike the study from Brazil, which has documented better treatment outcomes in those who received nutritional support [34,35]. Globally, the evidence on treatment outcomes following community-based nutritional support was inconclusive due to methodological limitations [33,34].

### Barriers and facilitators in the implementation of nutritional support

The donor-related barriers, like a shortage of supplies and challenges in sustaining their support, were similar to the findings from a Southern Indian study [15]. Also, the donors desired recognition and were more inclined to distribute food baskets within their geographical settings. As the majority of the *Ni-kshay Mitras* in Puducherry were individual *Mitras*, they preferred the distribution of food baskets to their nearest health facility. This poses a challenge in areas with no *Ni-kshay Mitras*. Barriers from the health providers' perspective were related to collecting the supplies from the donors and transporting them. Obtaining OTP based consent and reporting in the portal following the distribution was considered time-consuming. Amongst the adults with TB, issues pertaining to stigma and a sense of insufficiency with respect to the contents of the food baskets were observed and were similar to the findings from other studies [15,30].Concerns with respect to stigma were also reported in the India TB report 2024 [20].

The interventions that worked were the established protocol with the scheduled distribution once a month, coordination between the PHIs, honoring patients' feedback, and regular quality checks. The program managers felicitated donors and encouraged their participation. Suggestions included creating awareness through information, education, and communication (IEC) materials, prioritizing vulnerable groups for the distribution of food baskets, increasing the contents and variety in food baskets, and monitoring weight during the period of treatment.

**Strengths and limitations**

The study was novel in comprehensively exploring the coverage of nutritional support provided through the community-driven mechanisms to adults with TB in real-world settings. The insights from this pragmatic paradigm can guide the policy makers and program managers in making context-specific adaptations. The study included all adults with TB who received food baskets, which adds to the generalizability of the study findings to Puducherry. The gaps in secondary data and findings were explained in the qualitative phase by the representative sample of stakeholders. Finally, the data triangulation of the nutritional support coverage from multiple data sources adds to the robustness of our results. An inherent limitation persists due to the use of secondary data, as it may lead to misclassification of categories such as BMI, socioeconomic status, etc. We could not comment upon the details of the actual consumption of contents with the food basket, as no data was available. Also, we were not able to confirm this via weight gain due to missing data.

**Implications for policy and practice**

The study results were communicated to the program manager for routine monitoring and scaling up of this initiative. Indicators derived from this study, such as the proportion who consented among the notified adults with TB and those who received more than the median number of food baskets, could be used in routine program settings for monitoring purposes. IEC materials related to the *Ni-Kshay Mitra* initiative were developed by the study team and disseminated through the PHIs and health campaigns to increase the participation of donors. Posters and videos were created in the vernacular language for display at various PHIs and promotion through social media handles by the State TB cell.

**Recommendations**

Our findings highlight clear programmatic gaps and opportunities for strengthening the *Ni-kshay Mitra* initiative. The actual coverage (at least half the PwTB receiving three or more baskets) as compared to the expected (six baskets) indicates the need for real-time monitoring. The time interval of 41 days between diagnosis and first food basket highlights the need for aligning nutritional support with treatment initiation to reduce early mortality. A patient-centric model with the provision of food baskets as soon as the patient is linked to the current PHI should be established. There is a need for developing a mechanism to ensure the prior availability of food baskets from various donors in order to maintain the supply chain for need-based distribution. Equity analysis outlines the measures need to be undertaken in early initiation of nutritional support amongst the vulnerable, such as those with severe undernutrition (BMI < 16), with a target of achieving at least six food baskets following the standard of treatment. Donor-related challenges suggest the need to formalise donor engagement through reporting of patient outcomes and non-financial incentivisation (e.g., public recognition or tax benefits) for sustained support. Lastly, qualitative interviews from stakeholders suggest the request to increase food quantity and diversity, fostering the need for evidence-based food composition tailored to the needs of adults with TB.

## Conclusions

High consent rates and a diverse pool of donors in this initiative reflect commendable public participation in TB care, with nearly half of the adults with TB receiving at least three food baskets during their treatment period. The time interval from diagnosis to the receipt of the first basket was almost one and a half months. These indicators were similar across different socioeconomic and nutritional statuses. The evidence generated in this study calls for early initiation of food baskets among targeted groups (severely undernourished and below the poverty line) and sustaining donor engagement under routine programmatic settings. Insights from this study can inform the refinement and expansion of nutritional support initiatives across other regions.

## Supporting information

**S1 Text. Operational Definitions.**
(DOCX)

**S2 Text. Qualitative questionnaire and probes.**
(DOCX)

**S3 Text. Results of regression analysis.**
(DOCX)

**S1 Codebook. Codebook of Facilitators, barriers and suggestions.**
(XLSX)

**S1 GRAMMS Checklist. GRAMMS checklist.**
(DOCX)

## Acknowledgments

This operational/ implementation research that resulted in this manuscript was conducted through the Structured Operational Research and Training Initiative (SORT IT), a global partnership led by the Special Program for Research and Training in Tropical Diseases at the World Health Organization (WHO/TDR). The model is based on a course developed jointly by the International Union Against Tuberculosis and Lung Disease (The Union) and Medécins sans Frontières (MSF/Doctors Without Borders). This specific SORT IT course, which resulted in this publication, was part of year one of the ICMR-National Institute of Epidemiology (ICMR-NIE) led TB SORT IT course 2024–26, with support and guidance from India's Central TB Division and WHO India. It was jointly developed and implemented by: ICMR-National Institute of Epidemiology (ICMR-NIE), Chennai, India; ICMR-National Institute for Research in Tuberculosis (ICMR-NIRT), Chennai, India; Post Graduate Institute of Medical Education and Research (PGIMER), Chandigarh, India; FIND, New Delhi, India; Baroda Medical College, Vadodara, India; Narotam Sekhsaria Foundation, Mumbai, India; Government Medical College, Shahdol, India; All India Institute of Medical Sciences (AIIMS), Madurai, India; All India Institute of Medical Sciences (AIIMS), Bathinda, India; Yenepoya Medical College, Mangaluru, India; and GMERS Gotri Medical College, Vadodara, India. Gratitude to Dr. Vidhya Bharathi R and Dr. Vishva C for their support in obtaining secondary data from various PHIs of Puducherry. We appreciate the support of Ms. Elakia J, Technical content writer, Melange publication, Puducherry for proof reading the manuscript for grammatical errors and suggestions.

## Author contributions

**Conceptualization:** Revadi Gouroumourty, Kibballi Madhukeshwar Akshaya, Madhur Verma, Premanandh Kandasamy, Amol Dongre, Hemant Deepak Shewade.

**Data curation:** Revadi Gouroumourty, Swetha Mathivanan, Shephrine Wrobel R. Prasad, Venkatesh Couppoussamy.

**Formal analysis:** Revadi Gouroumourty, Amol Dongre.

**Funding acquisition:** Revadi Gouroumourty, Hemant Deepak Shewade.

**Investigation:** Revadi Gouroumourty, Swetha Mathivanan, Shephrine Wrobel R. Prasad, Venkatesh Couppoussamy.

**Methodology:** Revadi Gouroumourty, Kibballi Madhukeshwar Akshaya, Madhur Verma, Amol Dongre, Murugan Natarajan, Hemant Deepak Shewade.

**Project administration:** Revadi Gouroumourty, Premanandh Kandasamy, Swetha Mathivanan, Shephrine Wrobel R. Prasad, Venkatesh Couppoussamy.

**Resources:** Revadi Gouroumourty, Premanandh Kandasamy, Shephrine Wrobel R. Prasad, Hemant Deepak Shewade, Venkatesh Couppoussamy.

**Software:** Revadi Gouroumourty.

**Supervision:** Revadi Gouroumourty, Kibballi Madhukeshwar Akshaya, Madhur Verma, Premanandh Kandasamy, Amol Dongre, Hemant Deepak Shewade, Venkatesh Couppoussamy.

**Validation:** Revadi Gouroumourty, Kibballi Madhukeshwar Akshaya, Madhur Verma, Premanandh Kandasamy, Amol Dongre, Murugan Natarajan, Swetha Mathivanan, Shephrine Wrobel R. Prasad.

**Visualization:** Revadi Gouroumourty, Kibballi Madhukeshwar Akshaya, Madhur Verma, Amol Dongre.

**Writing – original draft:** Revadi Gouroumourty, Kibballi Madhukeshwar Akshaya, Madhur Verma, Amol Dongre, Swetha Mathivanan, Shephrine Wrobel R. Prasad.

**Writing – review & editing:** Revadi Gouroumourty, Kibballi Madhukeshwar Akshaya, Madhur Verma, Premanandh Kandasamy, Amol Dongre, Murugan Natarajan, Hemant Deepak Shewade.

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
