## [Decision Letter · Decision Letter 0]

24 Jun 2025

PGPH-D-25-01043

Exploring the Accessibility of Patients with Tuberculosis to Food Assistance Initiative: A Mixed Methods Study from Puducherry, India.

Dear Dr. Gouroumourty,

Thank you for submitting your manuscript to PLOS Global Public Health. After careful consideration, we feel that it has merit but does not fully meet PLOS Global Public Health’s publication criteria as it currently stands. Therefore, we invite you to submit a revised version of the manuscript that addresses the points raised during the review process.

We find the topic interesting and relevant. However, the manuscript needs improvement, especially in presentation of the methodology and result sections.

We look forward to receiving your revised manuscript.

Kind regards,

Sonali Sarkar

Academic Editor

Journal Requirements:

Additional Editor Comments (if provided):

Reviewers' comments:

Reviewer's Responses to Questions

**Comments to the Author**

1. Does this manuscript meet PLOS Global Public Health’s publication criteria?

Reviewer #1: Partly

Reviewer #2: Yes

2. Has the statistical analysis been performed appropriately and rigorously?

Reviewer #1: Yes

Reviewer #2: Yes

3. Have the authors made all data underlying the findings in their manuscript fully available (please refer to the Data Availability Statement at the start of the manuscript PDF file)?

Reviewer #1: Yes

Reviewer #2: Yes

4. Is the manuscript presented in an intelligible fashion and written in standard English?

Reviewer #1: No

Reviewer #2: No

Reviewer #1: The study focuses on nutritional status among TB patients, an important factor affecting the treatment outcomes. The write up has to be improved, for an easy and interesting read. Following suggestions can be considered to improve the quality of manuscript -

Title - accessibility as an outcome is not measured – has to removed from the write up

Abstract – balanced and whole some

Introduction – various outcomes of interest can be briefly justified,

novelty of the study remains questionable,

the expected outcome of improvement in nutrition among patient with TB can be included here.

Methods – using the Cohort study design is questionable – please include information on the two groups, follow up and assessment if retained as Cohort study

Elaborate on existing programme guidelines – about the collection and distribution of food baskets, for example various models of food basket delivery

It’s important to describe about Nikshay Poshan Yojana as it also addresses the nutritional needs of patients with TB

Mention the number of interviews conducted over phone

Elaborate qualitative data analysis - how was transcription done, time lag between interviews and transcription

Operational definition – include definitions of coverage and delay, equity of distribution

Is it a mixed method study, does the quantitative part answer any research questions like coverage or delay or just supports the qualitative data with description of available data from records.

Results - Figure 2 is not legible

Only 50 % received three food baskets – are the number of donors sufficient enough to provide food baskets to all the TB patients. The qualitative part should be strong to inform why only half of the eligibles received food baskets.

Qualitative results can be presented in a better way – suggestion: the quotes supporting the codes and themes can be presented alongside in the table in a separate column.

Equity in distribution and monitoring of weight gain – these outcomes are abruptly coming into the write up in the results section

Qualitative research was a good opportunity to explore challenges like – quality standards of the basket, whether the TB patients are actually consuming the food basket. If data is collected on the same, it can strengthen the discussion session

Reviewer #2: This paper provides new insights on the implementation aspects of Nikshay Mitra initiative of NTEP in Puducherry. Data and insights in the context of NM is rare and authors have taken the first step to fulfill this . While this attempt is commendable and they have brought in ample data and findings, still the manuscript requires major revisions as given below.

Title : Not clear and doesn’t reflect the full purpose of the study. Need to be edited.

Abstract :

The method section of abstract is vague . Needs to be edited based on comments given for the method part below.

Abstract Results should describe the specific themes of the qualitative study . Abstract Conclusion must only reflect based on the findings presented in the paper.

Methodology

Study design

Line 94-98 : Study design is mentioned as “explanatory mixed methods operational research involved a cohort study (quantitative phase)” which doesn’t make anything clearer. Better to mention this a mixed method study ( using secondary data for quantitative and IDI for qualitative)

Line 98 : What is meant (by pragmatic philosophy)? Clarify this? Is this related to the theoretical framework for qualitative component ?

Study population : The quantitative phase is based on secondary data and not based interviews with patients . Line 134-136 is giving a wrong perspective.

Line 145_ What is meant by extreme or deviant sampling? Do authors means positive deviance sampling ? Why it was chosen? PD approach requires very different frame work and sequential steps and authors miss to mention any of these

Data collection : Line 155-167 : On what basis the time period of secondary data was decided((1/4/2023-31/3/2024). Explain steps on how data quality was ensured? Missing data etc..Add a section quality control for both sets of data

Overall provide COREQ checklist needs to provided for methodology (https://onlinelibrary.wiley.com/pb-assets/assets/17416612/COREQ_Checklist-1556513515737.pdf).

Line : 197 Line Negative binomial mixed model regression to identify the predictors ? Line 260 : But why Poisson regression is mentioned in the result? Either way justify the use of the methods? Does outcome variable is skewed ?

Qualitative analysis methods needs to spelled out clearly ( COREQ needs to be followed) .

Results section :

Presenting only few broad themes is not appropriate for the qualitative analysis. Describe the framework uses, codebooks, sub themes themes etc in a flow chart. Describe saturation point.

Result should provide perspectives of different stakeholders ( MO, Frontline staff based on deviant category) which at present is missing

Findings of qualitative and quantitative data needs triangulation which at present is just spelled but not described in detail.

Recommendations section needs to be shortened. Don’t explain the follow up activities which is not related to the results . If so, then explain it as part of your study methodology ( under heading : participant engagement)

Conclusion must only reflect the findings presented in the paper ( Don’t mention the following Line 366-368 : We observed a higher proportion of adults with TB to have consented and subsequently 367 received nutritional support compared to previous Indian study and can be directly attributed 368 to administrative commitment]

The manuscript needs complete English edition and need to ensure quality of formatting ( especially the heading and sub heading needs to be done clearly which at present is very messy ) . Attach COREQ checklist mandatorily.

**Do you want your identity to be public for this peer review?** For information about this choice, including consent withdrawal, please see our Privacy Policy

Reviewer #1: No

Reviewer #2: **Yes: ** KARIKALAN NAGARAJAN

---

## [Decision Letter · Decision Letter 1]

2 Sep 2025

PGPH-D-25-01043R1

Food baskets for people with TB in Puducherry, India: How many, how early and who received?

Dear Dr. Gouroumourty,

Thank you for modifying the manuscript based on the suggestions from the reviewers. After careful consideration, we feel that it has merit but does not fully meet PLOS Global Public Health’s publication criteria as it currently stands. Therefore, we invite you to submit a revised version of the manuscript that addresses the points raised during the review process.

One of our reviewers has suggested some more changes to the manuscript by improving the language and also incorporating the programmatic implications, which can pave the way to enhancing the programme of distribution of food baskets to the TB patients.

We look forward to receiving your revised manuscript.

Kind regards,

Sonali Sarkar

Academic Editor

Journal Requirements:

Additional Editor Comments:

Reviewer #1: As noted by the reviewer, there is still a scope of improvement to make the manuscript more readable and improve the relevance by focusing on the programmatic aspects.

Reviewer #2: This reviewer is satisfied with the modifications.

Reviewers' comments:

Reviewer's Responses to Questions

**Comments to the Author**

Reviewer #1: All comments have been addressed

Reviewer #2: All comments have been addressed

publication criteria?

Reviewer #1: No

Reviewer #2: Yes

3. Has the statistical analysis been performed appropriately and rigorously?

Reviewer #1: Yes

Reviewer #2: Yes

4. Have the authors made all data underlying the findings in their manuscript fully available (please refer to the Data Availability Statement at the start of the manuscript PDF file)?

Reviewer #1: Yes

Reviewer #2: No

5. Is the manuscript presented in an intelligible fashion and written in standard English?

Reviewer #1: No

Reviewer #2: Yes

Reviewer #1: Manuscript Title: Food baskets for people with TB in Puducherry, India: How many, how early and who received?

Thank you for the revision. The manuscript is much stronger and detailed now. However, it demands clarity of ideas and more transparency in writing. The following suggestions can be considered to improve the quality of the manuscript –

Overall, the write-up has to be improved for English grammar. The entire study is largely descriptive and does not evaluate the programme.

Title and abstract –

The title specifies coverage, time, who received the basket while the objectives are about characteristics of Nikshay Mitras, models, coverage, time interval, facilitating factors and barriers of implementation. Enlisted objectives appear to be vague and vary significantly from each other. There is no coherence and findings may not contribute to the improvement of ongoing programme.

Methods

o Justify how the data on the characteristics of Nikshay Mitra contribute to this study

o Line 487 - Explain how Ni-Kshay Mitras were linked to the consented adults with TB within their geographical setting – is a challenge

o Line 120 – change in weight or change in weight gain. Add justification in the write-up why to discuss about weight gain, when there is no data available for the addressing the question.

o Rather than describing Quan and Qual under each heading, it would have been better to write it according to the sequence followed in conducting the study

o Avoid repetition of information in the manuscript. Example, specifying dates in multiple sentences. Also, there is a mismatch of dates specified in Figure 1.

o Table 2 – as the Kurt Lewin framework has been used, a better representation of data will be appreciated

o Line 96 – 105: The description about the pragmatic paradigm, appears inappropriate along with study designs. The suggestion is to discuss it as a strength of the study. Avoid the term retrospective cohort.

Results

o The entire section looks poorly structured. Instead of detailed description, I encourage using tables, graphs, diagram or models. Eg: simple table or bar chart showing distribution of timing (e.g., <2 weeks, 2–4 weeks, >4 weeks)

o Qualitative analysis – for many questions, only data has been provided in the form of quotes. There is no description of results. Eg: What are the results regarding the quality of the food basket

Discussion

o Line 391 – why there is an equity gap when – ‘observed that all adults with TB were distributed food baskets irrespective of their nutritional and socio-economic status since 2022.’ Line 407 and ‘These indicators were similar across different socio-economic and nutritional statuses.’ Line 473

o Line 403 - Avoid including multiple concept texts in a single sentence. Eg: what is the relevance of the time of receiving basket when the sentence is on the equity of distribution. It hinders the flow of reading and difficult to comprehend.

o Line 411 and 437 – Monitoring weight gain during treatment is mandatory. Failure to do so, shows the implementation gap. It should be discussed and cannot be included as a recommendation

o Implication of delay should be discussed – suggestion is to include the effect of undernutrition on TB outcome etc.

o Explicitly acknowledge the limitations – secondary data, incomplete data, risk of misclassification of socio-economic status etc.

Recommendation

o Make it more actionable by incorporating it to the existing programme. Eg: procuring the baskets one month prior etc.

Conclusion

o Avoid restating findings – add programmatic changes which are required to address the issue

Reviewer #2: The authors have addressed the comments given and the manuscript standard has been improved significantly.

**Do you want your identity to be public for this peer review?** For information about this choice, including consent withdrawal, please see our Privacy Policy

Reviewer #1: No

Reviewer #2: **Yes: ** Karikalan Nagarajan

---

## [Editor Report · Decision Letter 2]

27 Oct 2025

PGPH-D-25-01043R2

Coverage, Delivery models, and Implementation Challenges of the Community Driven Nutritional Supplementation Initiative for People with TB: a Mixed Methods Study from Puducherry, India

Dear Dr. Gouroumourty,

Thank you for submitting your manuscript to PLOS Global Public Health. After careful consideration, we feel that it has merit but does not fully meet PLOS Global Public Health’s publication criteria as it currently stands. Therefore, we invite you to submit a revised version of the manuscript that addresses the points raised during the review process.

The manuscript has improved after the modifications. However, some more language corrections are needed to convey the meaning clearly. Consistent use of terminologies are needed throughout the paper.

We look forward to receiving your revised manuscript.

Kind regards,

Sonali Sarkar

Academic Editor

Journal Requirements:

Additional Editor Comments (if provided):

Language of the manuscript has improved significantly. However, the following sentences still need to modified to convey the proper meaning. Please use terminologies uniformly across the manuscript. Both terms such as Ni-kshay and Nikshay have been used interchangeably.

Line 449: Heading needs to be changed to "Monitoring weight gain among those who received food baskets".

Line 471: The sentence "As the majority of the Nikshay Mitras in Puducherry were individual mitras, they prefer the distribution of food baskets to their nearest health facility" should be changed to "As the majority of the Nikshay Mitras in Puducherry were individual mitras, they prefered the distribution of food baskets to their nearest health facility".

Line 472: Sentence needs to be changed to "This poses a challenge in areas with no Ni-kshay mitras".

Line 513-16: The message in this sentence is not clear. Cosider the change to "The actual coverage (at least half the PwTB receiving three or more baskets) as compared to the expected (six baskets) indicates the need for real-time monitoring. The time interval of 41 days between diagnosis and first food basket highlights the need for aligning nutritional support with treatment initiation to reduce early mortality."
---

## [Editor Report · Decision Letter 3]

27 Nov 2025

Coverage, Delivery models, and Implementation Challenges of the Community Driven Nutritional Supplementation Initiative for People with TB: a Mixed Methods Study from Puducherry, India

PGPH-D-25-01043R3

Dear Dr Gouroumourty,

We are pleased to inform you that your manuscript 'Coverage, Delivery models, and Implementation Challenges of the Community Driven Nutritional Supplementation Initiative for People with TB: a Mixed Methods Study from Puducherry, India' has been provisionally accepted for publication in PLOS Global Public Health.

Best regards,

Sonali Sarkar

Academic Editor

All the suggestions from the reviewers have been addressed adequately. However, the point-of-contact for data access is one of the authors in the paper.